# A Stepwise Pharmacist-Led Medication Review Service in Interdisciplinary Teams in Rural Nursing Homes

**DOI:** 10.3390/pharmacy7040148

**Published:** 2019-11-05

**Authors:** Kjell H. Halvorsen, Torunn Stadeløkken, Beate H. Garcia

**Affiliations:** Department of Pharmacy, Faculty of Health Sciences, UiT The Arctic University of Norway, Tromsø N-9037, Norway; torunn_93@hotmail.com (T.S.); beate.h.garcia@uit.no (B.H.G.)

**Keywords:** nursing home residents, medication related problems, interdisciplinary team, clinical pharmacy

## Abstract

**Background:** The provision of responsible medication therapy to old nursing home residents with comorbidities is a difficult task and requires extensive knowledge about optimal pharmacotherapy for different conditions. We describe a stepwise pharmacist-led medication review service in combination with an interdisciplinary team collaboration in order to identify, resolve, and prevent medication related problems (MRPs). **Methods:** The service included residents from four rural Norwegian nursing homes during August 2016–January 2017. All residents were eligible if they (or next of kin) supplied oral consent. The interdisciplinary medication review service comprised four steps: (1) patient and medication history taking; (2) systematic medication review; (3) interdisciplinary case conference; and (4) follow-up of pharmaceutical care plan. The pharmacist collected information about previous and present medication use, and clinical and laboratory values necessary for the medication review. The nurses collected information about possible symptoms related to adverse drug reactions. The pharmacist conducted the medication reviews, identified medication-related problems (MRPs) which were discussed at case conferences with the responsible physician and the responsible nurses. The main outcome measures were number and types of MRPs, percentage agreement between pharmacists and physicians and factors associated with MRPs. **Results:** The service was delivered for 151 (94%) nursing home residents. The pharmacist identified 675 MRPs in 146 (97%) medication lists (mean 4.0, SD 2.6, range 0–13). The MRPs most frequently identified concerned ‘unnecessary drug’ (22%), ‘too high dosage’ (17%) and ‘drug interactions’ (16%). The physicians agreed upon 64% of the pharmacist recommendations, and action was taken immediately for 32% of these. We identified no association between the number of MRPs and sex (*p* = 0.485), but between the number of MRPs, and the number of medications and the individual nursing homes. **Conclusion:** The pharmacist-led medication review service in the nursing homes was highly successfully piloted with many solved and prevented MRPs in interdisciplinary collaboration between the pharmacist, physicians, and nurses. Implementation of this service as a standard in all four nursing homes seems necessary and feasible. If such a service is implemented, effects related to patient outcomes, interdisciplinary collaboration, and health economy should be studied.

## 1. Introduction

Nursing home residents have complex physical and mental health problems, and are often prescribed numerous of medications [1,2]. In Norway, more than 80% of nursing home residents have mild to severe cognitive impairments [3,4]. It is acknowledged that polypharmacy often leads to medication-related problems (MRPs), and is further associated with hospitalizations, morbidity, mortality, and a decline in quality of life [5,6]. Also, age-related changes in pharmacokinetics and pharmacodynamics increase the risk of MRPs in older people. The provision of rational medication therapy in frail elderly is a difficult task and requires extensive knowledge about medications [7,8]. Medication reviews in interdisciplinary teams have shown promising results to reduce MRPs in frail nursing home residents [9,10].

A medication review is “a structured evaluation of patient’s medicines with the aim of optimizing medicines use and improving health outcomes. This entails detecting medication-related problems and recommending interventions” [11]. The Norwegian Medicines Agency suggests that medication reviews should be performed at least once annually for all patients using four or more medications, or if the level of health care changes [12]. Medication reviews can be performed by physicians alone or in interdisciplinary teams with nurses and/or pharmacists according to Norwegian guidelines [13]. Although there are differences in how municipalities organize their health care services, we observe an increased involvement from pharmacists in interdisciplinary teams providing medication review services to nursing home residents.

Medication review studies conducted by interdisciplinary teams in urban areas report that MRPs are highly prevalent in nursing home residents and range from 1.8 to 5.1 per resident [10,14,15,16,17]. The most commonly identified MRPs concern unnecessary use of medications, excessive doses, the need for therapeutic monitoring, and risk of adverse drug reactions [10,14,15,16]. When it comes to medication reviews in rural areas, the literature is scarce.

### 1.1. Aim of the Study

We describe a stepwise pharmacist-led medication review service in combination with an interdisciplinary team collaboration in order to identify, resolve, and prevent MRPs in rural nursing homes residents. In addition, we describe outcomes of the interdisciplinary collaboration and explore differences in medication use and MRPs between the included nursing homes.

### 1.2. Ethics

The study was approved by the Regional Committee for Medical and Health Research Ethics and the Norwegian Centre for Research Data. Information about the study was provided both orally and written to the residents (or the next of kin of cognitive impaired residents). We received an oral informed consent from all residents or next of kin.

## 2. Methods

### 2.1. Setting and Study Population

This descriptive analytical study was performed from August 2016 to January 2017. In a defined rural area of Norway, we invited all nursing homes in six municipalities to participate in the pharmacist-led medication service. Four nursing homes from four municipalities accepted the invitation. The number of nursing home residents within each nursing home varied from 35 to 67.

### 2.2. Interdisciplinary Stepwise Approach

The stepwise approach comprised four steps: (1) patient and medication history taking; (2) systematic medication review; (3) interdisciplinary case conference; and (4) follow-up of pharmaceutical care plan, see Figure 1.

**Step 1.** Patient and Medication History Taking

The main purpose of the first step was for the pharmacist to get a comprehensive understanding of the patients’ current health status and medication therapy. This included collecting information on past and present medical history and medication use, including any present complaints or potential adverse drug reaction symptoms (e.g., dryness in mouth, cognitive impairment). For each resident, the pharmacist scrutinized medical records, medication charts, and laboratory values with a special focus on ongoing diseases and symptoms. Recent reports from physicians, nurses, or auxiliary nurses concerning aspects related to the use of medications was also considered, e.g., if medications used as needed were frequently administered or caregivers were facing administering problems.

The following data was collected for each resident: age, sex, weight, diseases, medications (regular and as needed), dosages and strengths and clinical and laboratory values relevant for medication reviews.

Medication use was classified as either no polypharmacy (<5 medications), polypharmacy (5–9 medications), or hyperpolypharmacy (10+ medications), and according to WHO’s anatomical therapeutic chemical (ATC) classification system [18].

**Step 2.** Systematic Medication Review

The objective of the second step was to identify MRPs among the nursing home residents, followed by recommendations on how to resolve or prevent them, i.e., develop a proposal for a pharmaceutical care plan for each resident. The pharmacist applied validated tools to facilitate and standardize the process. In this study, the following ten risk areas were included: (1) medications requiring therapeutic drug monitoring; (2) potential inappropriate medications for elderly; (3) problems related to administration or dosage forms; (4) drug interactions; (5) dosages or medications not suitable for the individual patient (e.g., due to renal or liver failure); (6) lacking indication for drug therapy; (7) inappropriate length of therapy for temporarily used medications; (8) suboptimal treated or untreated diagnosis or symptoms; (9) medications causing adverse drug reactions or change in clinical and laboratory values; and (10) need for monitoring of treatment. In addition, the screening tool of older people’s prescriptions (STOPP) version 2 and the Norwegian General Practice Nursing Home (NORGEP-NH) criteria were applied to identify potentially inappropriate medications, and the screening tool to alert to right treatment (START) version 2 to identify potentially omitted medications [19,20]. Additional tools applied during the medication review included drug interaction databases [21,22], clinical therapy guidelines available from the Norwegian Electronic Health library [23], the Norwegian Drug and Therapeutic Formulary for Health Personnel [24], and the Norwegian Pharmaceutical Product Compendium [25]. Glomerular filtration rates (GFR) were estimated by calculating creatinine clearance applying both the Cockcroft–Gault and the CK–Epi formulas [26].

Based on this comprehensive assessment we collected information related to number and types of MRPs identified and medications involved. Identified MRPs were classified according to the Norwegian classification system [27] comprising six main categories: (1) Drug of choice (a. need for additional drug, b. unnecessary drug, or c. inappropriate drug); (2) Dosage (a. too high, b. too low, c. suboptimal time of dosage, d. suboptimal formulation); (3) Adverse drug reactions; (4) Drug interactions; (5) Deviant drug use (a. by health personnel or b. by patient); and (6) Other (a. need for or lacking monitoring, b. unclear documentation in medication chart or prescription or c. other).

**Step 3.** Interdisciplinary Case Conference

In this step, health care professionals responsible for the nursing home residents, i.e., the physician, nurses, auxiliary nurses, and the pharmacist met at the respective nursing homes. The pharmacist presented the identified MRPs which were discussed with the other team members who also could present additional MRPs. The team reached consensus about actions, priorities, and individual pharmaceutical care plans including interventions to resolve or prevent the identified MRPs. The physician remained responsible for the medication therapy and had the final word in case of disagreement.

**Step 4.** Follow-Up of Pharmaceutical Care Plan

In this final step, the health care team followed-up the agreed-upon individual pharmaceutical care plans, either immediately or after the case conference.

To describe outcomes of the pharmacist-led medication review service and the interdisciplinary team collaboration, we combined the outcomes from Step 3 and 4. These were categorized as following: (1) interdisciplinary team agreed, action taken immediately; (2) interdisciplinary team agreed, action unclarified; (3) interdisciplinary team disagreed; (4) problem solved before case conference; (5) interdisciplinary team agreed, patient disagreed; (6) independent pharmacist intervention; and (7) patient deceased before case conference.

### 2.3. Statistical Analysis

Data collected during each of the abovementioned steps were used in descriptive analysis; medication use (frequencies and type, STEP 1), MRPs (frequencies and type, STEP 2), and agreement within the interdisciplinary team (frequencies and type, STEP 3).

We present continuous variables as means with standard deviation (SD) and categorical variables as percentages. We applied Student’s *t*-test and Chi-square for normally distributed data. For nonparametric data, we applied the Kruskal–Wallis test and the Mann–Whitney test. We conducted a Pearson product–moment r correlation to assess the relationship between number of MRPs and number of medications and sex. A linear regression analysis was conducted to investigate association between number of MRPs (dependent variable), and number of medications, sex, and nursing homes. We also conducted an automatic regression modeling analysis to identify predictor factors (PFs) associated with the numbers of MRPs (dependent variable), and to assess to which degree the model could explain the variance in number of MRPs. *P* values < 0.05 were considered statistically significant. We used Microsoft Excel and IBM SPSS Statistics 24 for Windows for data management and analysis.

## 3. Results

**Step 1.** Patient and Medication History Taking

We included 151 of 160 (94%) nursing home residents (67.5% women, mean age 85.2 years, range: 52–100 years), see Table 1. Nine residents were not included due to no willingness to participate (n = 2), not able to provide informed consent (n = 5), not at the nursing home at the time of medication review (n = 1), or receiving intensive palliative care (n = 1).

The nursing home residents used in total 1793 medications (regular and as needed); altogether 220 different active ingredients. Total medication use per resident was 11.9 (mean, SD 4.4, range 1–27); for regular use 8.0 (mean, SD 3.4, range 1–18) and as needed 3.7 (mean, SD 2.2, range 0–11), see Table 1. Hyperpolypharmacy was present in 34% of nursing home residents.

The mean number of medications per resident varied significantly between the nursing homes (range: 10.6–13.1, *p* = 0.021), so did the number of medications used regularly (range: 6.0–9.3, *p* < 0.001) and medications used as needed (range: 2.8–4.5 medications, *p* = 0.008).

The most frequently medications used regularly were paracetamol (70.2%), cholecalciferol (39.1%), lactulose (36.4%), vitamin B12-complex (33.8%), and escitalopram (25.8%); while oxazepam (52.3%), paracetamol (39.1%), oxycodone (21.8%), acetylcysteine (17.9%), and tramadol (17.9%) were the most frequently medications used as needed.

We also identified prescribing differences between nursing homes. For instance, one nursing home dispensed laxatives regularly as standard treatment to all residents every morning, which was not indicated in the medication chart. Another example was differences with regard to chronic pain management, where buprenorphine was the standard medication prescribed in one nursing home, while fentanyl was prescribed in the other three.

**Step 2.** Systematic Medication Review

The pharmacist identified in total 675 MRPs (mean 4.6 MRPs per resident, SD 2.6, range 0–13) in 146 (97%) of the nursing home residents. The MRPs most frequently identified concerned ‘unnecessary drug’ (22.2%), ‘too high dosage’ (16.7%), and ‘drug interactions’ (16.4%), see Figure 2. The mean number of identified MRPs per resident varied significantly between the nursing homes, from 2.7 (SD = 1.9) in nursing home 1, to 5.7 (SD = 2.4) in nursing home 3 (*p* < 0.001). We identified no relationship between the number of MRPs and sex (*p* = 0.485).

**Step 3–4.** Interdisciplinary Case Conferences and Follow-Up on Pharmaceutical Care Plan

A total of 563 (83%) MRPs were discussed within the interdisciplinary teams. The remaining 112 MRPs (17%) had either been resolved before the interdisciplinary case conference (n = 82), the pharmacist had performed an independent intervention (n = 1), or patients had died before the case conference (n = 29). The interdisciplinary teams agreed to pharmacist recommendations for 63.2% of the MRPs presented, where 32.3% were resolved during the case conference. The interdisciplinary teams mainly agreed upon recommendations concerning unclear documentation, drug interactions, suboptimal dosing regimen, monitoring required, and too high dosage (>50% acceptance rate), see Figure 2. The interdisciplinary teams most often disagreed upon recommendations concerning adverse drug reactions, inappropriate drug choice, and unnecessary drug (<30% acceptance rate). The level of agreement varied between the nursing homes, from 59.6% in nursing home 2 to 72.7% in nursing home 3.

The number of MRPs per resident was significantly correlated with the number of regularly used medications (Pearson r = 0.59, *p* < 0.001), but not with age (r = 0.05, *p* = 534) or sex (r = −0.069, *p* = 0.406). Results from the linear regression analysis showed that the number of identified MRPs per resident was positively associated with the number of medications per patient (β = 0.653, *p* < 0.001) and the nursing homes (β = 0.223, *p* < 0.002, R^2^ = 0.452).

The automatic regression modeling analysis identified several factors that were significantly associated with the number of MRPs per residents; use of zopiclone, use of oxazepam, number of regularly used medications (PF = 0.68, *p* < 0.001), and nursing home (PF = 0.26, *p* < 0.001). The latter two factors combined explained 45.6% (R^2^ = 0.456) of the variance of the dependent variable.

## 4. Discussion

This study demonstrates that a stepwise pharmacist-led medication review service in interdisciplinary teams is feasible in order to identify, resolve, and prevent MRPs in rural nursing home residents. Based on clinical information, followed by a structured and comprehensive medication review, the interdisciplinary team agreed to a high proportion of the identified MRPs and recommendations. Despite being aware that nursing home residents often use unnecessary medications, too high doses, and medications with potential for drug interactions, the interdisciplinary team disagreed mostly concerning the pharmacist recommendations related to adverse drug reactions, inappropriate drug choice, and unnecessary drug use.

### 4.1. Medication Use and Medication Related Problems

In total, more than 220 different pharmacological agents were prescribed to the nursing home residents; resulting in an average use of 11.9 medicines. This is consistent with other recently published Scandinavian studies reporting an average of 11.5–11.9 medications per resident [10,28]. We observed significant differences between the nursing homes with regards to mean number of medications used per resident. This can partly be explained by differences in physicians prescribing habits and what they include in medication charts.

The comprehensiveness of the interdisciplinary stepwise model enabled identification of a wide range of MRPs in almost all residents; many of them with the potential to worsen health conditions considerably (see Figure 2). Due to the use of different MRP classification systems, it is challenging to make sound comparisons with other studies. However, in relation to type, i.e., unnecessary use of medication (and number of MRPs), our findings from this rural area are consistent with former studies from Norway, Sweden, Switzerland, and the USA [10,29,30,31,32]. In a recent study using the same classification tool as we did, Devik et al. found a mean of 3.7 MRPs per nursing home resident. They found that most of the MRPs related to unnecessary medication use (28%) or to need for additional medications (22%), while we identified most problems in relation to unnecessary medication use, to high doses and drug interactions [9].

In light of recent initiatives, like the Norwegian patient safety campaign focusing on correct use of medicines in nursing homes [33], and the growing body of literature promoting deprescribing among these residents [34], one should expect a decline in both MRPs and medication use. Although differences in type of MRPs identified, due to variance in disease status, symptoms, and comorbidities among the included residents, these findings combined calls for attention when it comes to the use of unnecessary medications. Moreover, they confirm the necessity to adjust ongoing strategies, and focus more on building interdisciplinary team capacity that can maintain and secure rational use of medication in these individuals.

### 4.2. The Interdisciplinary Team Collaboration

The method applied in this study descends from evidence advocating that pharmacist-led medication review services in combination with interdisciplinary teams enable identification and reduction of MRPs among nursing home residents and consequently improved quality of medication prescribing [6]. Recently, National Health Service (NHS) England announced that “thousands of pharmacists and pharmacy technician will be sent into care homes to carry out checks on residents, reviewing their medicines” [35]. In Norway, it is still rather uncommon that pharmacists are engaged as regular primary care interdisciplinary team members. This might be due to the fact that many of the pharmacist-led medication review studies conducted in nursing homes have been purely descriptive cross-sectional, or before and after studies [9,10]. Consequently, they have failed to provide strong evidence about how identifying, solving and preventing MRPs may affect endpoints like quality of life, hospitalization, and death.

The interdisciplinary team agreed upon 62% of the MRPs identified by the pharmacist, which is lower, but within range of the acceptance rate of 71% reported by Devik et al. [9]. The purpose of the MRP classification tool was to investigate areas of sub-optimal quality in relation to the residents’ entire medication regimen. Even though the classification tool has not been validated in terms of the abovementioned endpoints, the pharmacist recommendations combined with acceptance rates by the interdisciplinary teams could be considered reasonable proxy endpoints. We believe these endpoints provide insightful information related to the clinical relevance of the MRPs and the quality of medication therapy delivered for this specific population [36].

One of the main advantages of presenting and discussing MRPs within an interdisciplinary team is the opportunity to conduct prompt adjustments. Almost one-third of all identified MRPs where taken care of immediately during the case conference. Several of these concerned lowering the dose of or discontinuing medications; both interventions with potential to lower drug burden and reduce the risk of adverse drug reactions. It is also beneficial to discuss the clinical relevance of the identified MRPs on an individual level, and even more importantly to learn from each other’s professional conducts.

### 4.3. Factors Associated with MRPs

The automatic regression modelling identified two factors that combined explained almost 50% of the variance in the numbers of MRPs, i.e., number of medications and the nursing homes. Previous studies investigating MRPs in hospital patients have found a similar linear correlation between MRPs and number of medications used per patients [37]. A number of factors, including available staff resources and differences in skills, knowledge, and attitudes among them, could cause variance in MRPs between the nursing homes. Interestingly, two of the nursing homes in the present study had been involved in the Norwegian patient safety campaign focusing on appropriate use of medication. We anticipated that this would be recognizable in terms of numbers of medication used or MRPs, but it was not.

### 4.4. Strengths and Limitations

The fact that the interdisciplinary teams agreed to more than 60% of the pharmacist recommendations, underpin that the quality of medication administration in this population is still of concern. The comprehensive data collection and broad interdisciplinary knowledge of the residents’ clinical status, allowed the interdisciplinary team to conduct reliable clinical assessments. Besides, the systematic methodology and the use of valid tools such as NORGEP-NH, START, and STOPP facilitated the interdisciplinary team to reassess ongoing medication therapy and conducting clinically sound decisions.

Limitations include a small study population in a rural part of Norway and that only one pharmacist performed the medication reviews. Also, the study design was purely descriptive, and do not include a control group. Furthermore, the stepwise medication review and the interdisciplinary collaboration process has not been validated. In addition, this was the first time that the team members experienced to collaborate with a pharmacist and vice versa, which could have influenced the discussions around MRPs and acceptance rates. It is fully acknowledged that it requires time to build trust and establish well-functioning interdisciplinary teams that overcome barriers that impede interdisciplinary team work [38].

## 5. Conclusions

This study elaborates how a stepwise pharmacist-led medication review, in close collaboration with an interdisciplinary team facilitates identification, resolving, and prevention of MRPs in nursing home residents. The frequency of MRPs identified is similar to previous studies, and is still associated with the total number of medications used. Based on the high level of agreement within the interdisciplinary team in this study, we believe that nursing homes residents in general may benefit from a closer collaboration with clinical pharmacists. Our results demonstrate that this service should be delivered on a regular basis, in close collaboration with the nursing home physicians and nurses. However, to better understand the benefits of interdisciplinary teamwork, health politicians should decide upon appropriate outcome measures, i.e., quality indicators that more objectively can be applied to measure and validate the added-value of this service.

## Figures and Tables

**Figure 1 pharmacy-07-00148-f001:**
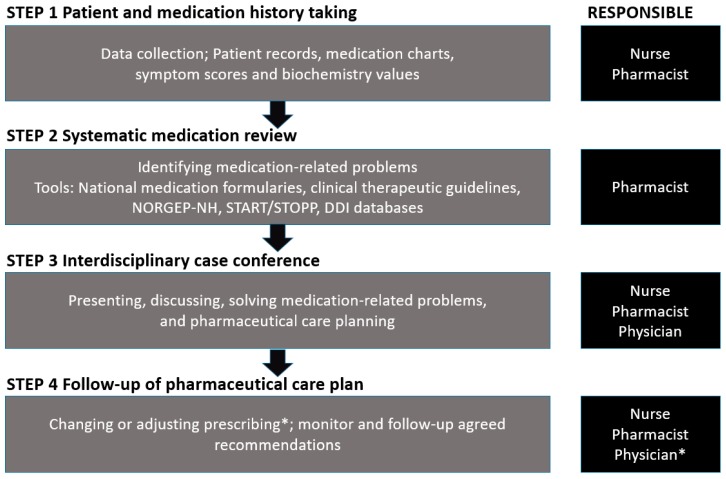
Pharmacist-led interdisciplinary medication review service.

**Figure 2 pharmacy-07-00148-f002:**
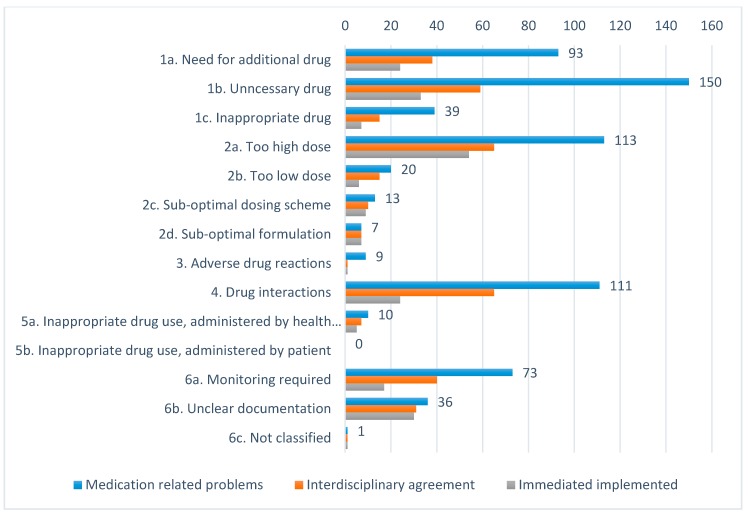
Distribution of types of medication related problems (MRPs), interdisciplinary agreement and immediately implemented interventions.

**Table 1 pharmacy-07-00148-t001:** Demographic and clinical variables of the included nursing home residents (n = 151).

Nursing Home		1		2		3		4		Total	
		n	SD (%)	n	SD (%)	n	SD (%)	n	SD (%)	n	SD (%)
**Participants**		35		43		31		42		151	
	Female	23	(65.7)	31	(72.0)	20	(64.5)	28	(66.7)	102	(67.5)
	Male	12	(34.3)	12	(28.0)	11	(35.5)	14	(33.3)	49	(32.5)
**Age, years**											
	Mean	83.1	10.1	86.4	6.9	84.4	8.8	86.4	8.4	85.2	8.6
**Mean months in nursing home**		28.9	26.1	36	32.3	20.4	25.3	40.8	70.5	32.5	44.5
**Mean number of medications**											
	Total	11.1	3.3	13.1	4.5	12.9	5.5	10.6	4.0	11.9	4.4
	Used regularly	8.2	2.8	8.9	3.1	9.3	3.6	6	3.2	8.0	3.4
	Used as needed	2.8	1.4	3.8	2.1	3.5	2.6	4.5	2.4	3.7	2.2
	Short course	0.03	0.2	0.37	0.6	0.1	0.4	0.19	0.5	0.19	0.5
**Mean number of MRPs**		2.7	1.9	5.2	2.6	5.6	2.4	4.2	2.6	4.4	2.6
**Mean weight, kg**		71.2	15.2	66.7	15.8	65.6	13.9	66.9	12.7	67.6	14.5
**Mean systolic blood pressure, mmHg**		134.3	19.6	134	24.9	136.3	20.7	134.7	17.9	134.7	20.9
**Mean diastolic blood pressure, mmHg**		75.4	10.1	73.2	14.2	79.9	17.8	73.5	10.3	75.2	13.4
**Mean eGFR, CKD-EPI mL/min**		70.1	38.5	60.2	17.2	65.4	22.3	64.6	20.1	64.9	25.4
**Mean eGFR, Cockcroft Gault mL/min**		66.6	58.5	50.0	21.1	50.5	18.3	53.7	23.6	55	34.2
**Polypharmacy categories**											
	0–4 medicines, no polypharmacy	1	(3)	1	(2)	4	(13)	15	(36)	21	(14)
	5–9 medicines, polypharmacy	21	(60)	24	(56)	12	(39)	21	(50)	78	(52)
	10+ medicines, hyperpolypharmacy	13	(37)	18	(42)	15	(48)	6	(14)	52	(34)

Abbreviations; SD = standard deviation; MRPs = medication-related problems; kg = kilogram; mmHg = millimeter mercury; mL/min = milliliters per minute; eGFR = estimated glomerulus filtration rate; CKD-EPI = chronic kidney disease epidemiology collaboration

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
