# Peer review of "A Stepwise Pharmacist-Led Medication Review Service in Interdisciplinary Teams in Rural Nursing Homes"

_pharmacy, 2019, doi:10.3390/pharmacy7040148_

Round 1

Reviewer 1 Report

Abstract

Line 17 drug-related problems (MRPs) – it is ideal to be consistent with terms to avoid confusion The results section mentions several proportions, but it is unclear what the denominator is. It would be ideal to mention total number of people in the study cohort and use that as a denominator. How was the associated between MRPs and sex, number of medicines and nursing homes measured?

Figure 1 – define abbreviations used

Methods

Step 1 – author suggests they ‘will’ collect information – please amend

Results

Did the total number of medicines include medicines prescribed/used on a ‘prn’ basis as well? Line 160: Please define ‘active substances’, does the author mean active ingredient?

Discussion

Lines 218-219: this is new result which should have been presented in the results section rather than discussion

Author Response

Abstract

Line 17 drug-related problems (MRPs) – it is ideal to be consistent with terms to avoid confusion

Authors' response: Changed.

The results section mentions several proportions, but it is unclear what the denominator is. It would be ideal to mention total number of people in the study cohort and use that as a denominator.

Authors' response: We have added information about the denominator, i.e., residents when appropriate. However, in relation to MRPs, we have used the total number of MRPs as denominator as we think this better reflect what type of problems that the team needed to deal With.  

How was the associated between MRPs and sex, number of medicines and nursing homes measured?

Authors' response: We have stated in the method section that "we conducted a Pearson product-moment r correlation to assess the relationship between number of MRPs and number of medications and sex. A linear regression analysis was conducted to investigate associations between number of MRPs (dependant variable), and number of medications, sex and nursing homes.

Figure 1 – define abbreviations used

Authors' response: We have now defined the different abbreviations. 

Methods

Step 1 – author suggests they ‘will’ collect information – please amend

Authors' reponse: Thanks for the comment; we have rewritten the paragraph to past tense. 

Results

Did the total number of medicines include medicines prescribed/used on a ‘prn’ basis as well? Line 160: Please define ‘active substances’, does the author mean active ingredient?

Authors' response:  We think this information is included, please also see table 1. We have change "substances" to "ingredients"- thanks for pointing this out. 

Discussion

Lines 218-219: this is new result which should have been presented in the results section rather than discussion

Authors' response: Thanks for the comment - we have moved this section to the result chapter. 

Reviewer 2 Report

The manuscript is based on a sound premise, a significant foundation of prior work, and is innovative. The step-wise collaborative approach is very innovative and is a strength of this paper. The paper is easy to follow and well-written. The following suggestions are for the authors' consideration:

1. Figures 1 and 2 are hard to read and could use some editing to make them easier to read.

2. Methods: Data collection and data analysis are described within the four steps of the study. This is effective, but I wonder if there should be more detail that would further describe the overall aim into sub-aims that would link to each step of the study.

3. Related to comment 2 above, measures used for data collection could be described in more detail and then linked to each step of the study. Perhaps an overall summary at the end of the methods section would be an effective way to do this.

These are just suggestions for the authors' consideration. A strength of this paper is the step-by-step flow of the paper. Please keep that in mind as you consider the three suggestions given above. 

Author Response

Thanks for Your valuable comments and review. Please find our response below: 

The manuscript is based on a sound premise, a significant foundation of prior work, and is innovative. The step-wise collaborative approach is very innovative and is a strength of this paper. The paper is easy to follow and well-written. The following suggestions are for the authors' consideration:

1. Figures 1 and 2 are hard to read and could use some editing to make them easier to read.

Authors' repsonse: We have made adjustments to the figures, and hopefully this will improve the readability. 

2. Methods: Data collection and data analysis are described within the four steps of the study. This is effective, but I wonder if there should be more detail that would further describe the overall aim into sub-aims that would link to each step of the study.

3. Related to comment 2 above, measures used for data collection could be described in more detail and then linked to each step of the study. Perhaps an overall summary at the end of the methods section would be an effective way to do this.

Authors response to comments 2 and 3: This is a good suggestion; we have therefore in the beginning of the Statistical analysis (end of Method section) included the following paragraph: 

"Data collected during each of the abovementioned steps were used in descriptive analysis; medication use (frequencies and type, STEP 1), MRPs (frequencies and type, STEP 2) and agreement within the interdisciplinary team (frequencies and type, STEP 3)."

These are just suggestions for the authors' consideration. A strength of this paper is the step-by-step flow of the paper. Please keep that in mind as you consider the three suggestions given above. 

Authors response: We have also re-read the entire manuscript and conducted minor misspellings, and English Language. 

Round 2

Reviewer 1 Report

None